# Numerical Simulation and Experimental Study on Energy Absorption of Foam-Filled Local Nanocrystallized Thin-Walled Tubes under Axial Crushing

**DOI:** 10.3390/ma15165556

**Published:** 2022-08-12

**Authors:** Wei Wang, Yajing Wang, Zhen Zhao, Zhenzhen Tong, Xinsheng Xu, Chee Wah Lim

**Affiliations:** 1State Key Laboratory of Structural Analysis for Industrial Equipment, Department of Engineering Mechanics, Dalian University of Technology, Dalian 116024, China; 2Dalian Metro Co., Ltd., Dalian 116011, China; 3College of Locomotive and Rolling Stock Engineering, Dalian Jiaotong University, Dalian 116028, China; 4Key Laboratory of Impact and Safety Engineering, Ministry of Education, Ningbo University, Ningbo 315211, China; 5Department of Architecture and Civil Engineering, City University of Hong Kong, Hong Kong SAR, China

**Keywords:** aluminium foam, axial loading, crashworthiness, energy absorption, local surface nanocrystallization, thin-walled tube

## Abstract

A crashworthiness design of foam-filled local nanocrystallized thin-walled tubes (FLNTs) is proposed by using foam-filled structures and ultrasonic impact surface treatment. The crashworthiness and deformation modes of FLNTs are studied using an experiment and numerical analysis. A finite element numerical model of FLNTs is established, and the processing and test platform of FLNTs is set up to verify the numerical predication and analytical design. The results show that local nanocrystallization is an effective method to enhance crashworthiness for hexagonal FLNTs. The FLNTs with four circumferential continuous stripes of surface nanocrystallization exhibit a level of 47.12% higher specific energy absorption than the untreated tubes in numerical simulations for tubes with a 50% ratio of nanocrystallized area. Inspired by the strength mechanism, a novel nested foam-filled local surface nanocrystallization tube is further designed and studied in detail.

## 1. Introduction

Thin-walled structures have been widely used in passive vehicle safety systems as among the most typical crash energy absorber elements due to their light weight and high-energy absorbing capacity. Significant efforts have been taken to improve the energy absorbing ability of thin-walled structures under axial crushing. In recent decades, numerous studies were carried out to investigate the energy absorption capacity enhancement by experiments and numerical analyses [1,2,3,4,5,6,7,8,9]. In engineering devices, energy absorbers are designed to absorb the largest impact energy with the smallest mass. In this regard, foam filling has become an effective way for improving energy absorption of thin-walled structures because of the material and structural interaction between wall and foam core. It is therefore very promising to design new energy absorbers by using foam-filled thin-walled tubes.

Recently, much attention has been paid to foam-filled thin-walled structures. A large number of research works have been conducted to study the influence of aluminum foam on energy absorption of thin-walled structures. Meguid et al. [10,11,12,13,14,15] studied the axial collapse behavior of various foam-filled structures and revealed the effect of key influencing parameters on the collapse mechanism and energy characteristics. Goel et al. [16] conducted the deformation and energy absorption studies on a multi-wall tube structure with an aluminum core. They concluded that the multi-wall tube structure with foam alters the deformation modes and it results in a substantial increase in energy absorption capacity in comparison with a single or multi-wall tube without a foam core. Hanssen et al. [17] conducted experiments on the energy absorption characteristics of circular aluminum columns filled with aluminum foam under both quasi-static and dynamic loading conditions. They reported that the extrusion wall interaction effect caused an increase in the mean load while foam-filled tubes showed less dependency on load conditions. Mirfendereski et al. [18] conducted a series of parametric studies on foam-filled tapered tubes and they found that tapered rectangular foam-filled tubes subjected to axial static and dynamic loading conditions were more influenced by foam density. Chen et al. [19] carried out analytical and numerical studies on energy absorption characteristics of single-cell and multi-cell foam-filled structures. They found that the interaction effect between foam and column wall caused higher specific energy absorption. An et al. [20] proposed a foam-filled thin-wall tube with functionally lateral-graded thickness (FLGT) for accommodating both axial crushing and lateral bending. The numerical and optimal study showed that foam-filled FLGT structures have advantage over the traditional foam-filled uniform thickness (UT) structures under both axial crushing and lateral bending conditions. Yu et al. [21] experimentally studied the energy absorption of aluminum foam-filled tubes with different cell structures under compression and they concluded that aluminum foams of higher density contributed to the increase in energy absorption by an interaction effect. Salehi et al. [22] experimentally investigated the deformation modes and energy absorption of functionally graded metallic foam-filled tubes under impact loading and the result showed that the Al-A356 foam-filled tube was the best lightweight crashworthy structure. Zhang et al. [23] proposed a foam-filled tube of bio-inspired density and the result indicated a further level up to 24% higher energy absorption than the uniform foam-filled tubes. Song et al. [24] presented a bio-inspired tube with foam filler based on straw structures and their results showed that a design of four-square holes and four tapered holes has significant energy absorption capacity. Moreover, Song et al. [25] proposed a hat sectional (top hats and double hats) thin-walled structure with foam filler. They found that the energy absorption was increased both in the hat section and the foam core when filled with foam. The interaction effect was mainly contributed by the densification effect of aluminum foam. Using experiments and numerical analyses, Zhang et al. [26] investigated the crashworthiness for self-lock multi-cell tubes with foam filler. They found that a combination of aluminum foam filler and square tube envelope was the most effective design compared to the expanded polystyrene foam filler. Hou et al. [27,28] carried out multi-objective optimization of square and tapered circular foam-filled structures and they further improved the crashworthiness of the structures. Sun and his co-workers [20,29,30,31,32,33,34] studied crashworthiness for a series of foam-filled thin-walled structures and carried out multi-objective optimization for the structures. In addition, the crashworthiness for thin-walled structures filled with different foam fillers, for examples, polyurethane foam [35,36,37,38], polyvinyl chloride foam [39], auxetic foam [40,41], liquid nanofoam [42,43], etc., was further studied.

In the existing literature, thin-walled energy absorbers are mainly fabricated by steels due to their excellent mechanical properties and low costs. Therefore, it is difficult to improve their energy absorption capacities if there is no change in geometry and material. Fortunately, with the development of nanotechnology, surface nanocrystallization techniques have been proposed to improve the mechanical and chemical properties of materials. The principle of surface nanocrystallization involves making the surface of bulk metals or subsurfaces with severe plastic deformation by applying a load such that the grains are gradually refined to form nanograins [44]. Lu and Lu [45] first proposed a process of surface nanocrystallization by surface mechanical attrition treatment (SMAT) and shot peening treatment. Material grains can also be refined by angular extrusion [46], high-pressure torsion [47,48], ultrasonic impact treatment [49,50], ultrasonic shot peening treatment [51,52], etc. Lu et al. [53] found that SMAT can significantly increase the yield strength of 304 stainless steel, and the fully SMATed energy absorbing device was manufactured and designed to capitalize its excellent mechanical properties. This design can absorb more impact energy if it is compared with the existing Toyota Yaris energy absorbing device. Xu et al. [54] studied a local surface nanocrystallized energy-absorbing structure and they showed that it has 64.29% higher energy absorption than an ordinary tube. Meanwhile, they found that the model with three circumferential staggered stripes is an optimal design, and the ratio of local nanocrystallization was 50%.

Currently the main energy absorbers are thin-walled structures with square sections. Researchers [55,56,57,58,59] investigated axial crushing behaviors for cylindrical shells through theoretical analyses. Some available studies demonstrated that for axial crashing, hexagonal tubes are more appropriate for energy absorption than the others with polygonal shapes [60]. As a result, regular hexagon tubes are adopted to achieve the research objective in this study. Xu et al. [54] demonstrated that local surface nanocrystallization can significantly enhance energy absorption of thin-walled structures without the expense of an increased peak crushing force. However, the crashworthiness of foam-filled local surface nanocrystallized thin-walled structures with hexagonal section and the interaction effect between local nanocrystallized wall and foam have not been studied. In this study, aluminum foam filling and local surface nanocrystallization are combined to design a type of lightweight energy absorption device with better crashworthiness.

For the above reasons, a foam-filled local surface nanocrystallized hexagonal thin-walled tube is proposed for axial crashing energy absorption. The technique of ultrasonic impact treatment (UIT) is employed to generate surface nanocrystallization with a 50% proportion of treated area in the structure. The material properties for thin-walled structures and aluminum foam are obtained by testing. Subsequently, the energy absorption capacity of foam-filled local nanocrystallization tube is carried out using an experiment and the interaction effect is analyzed. A finite element (FE) model is established to analyze the foam-filled local nanocrystallized tubes under axial crushing with different stripe numbers, with a 50% ratio of nanocrystallization area. The interaction effect of foam-filled local UIT tube is significantly enhanced compared with the untreated tubes. In addition, a novel nested UIT tube with foam-filled is designed and studied based on the strength mechanism.

## 2. Experimental Tests

### 2.1. Ultrasonic Impact Treatment

The UIT equipment is assembled on a three-dimensional platform for producing nanocrystallization patterns in an efficient manner. Ring columns with different masses are attached to the UIT head to enhance the effect of UIT and to avoid the waggle of impact head during processing. The impact head is made of W18Cr4V high speed steel, which can bear repeated impact at high frequencies. The UIT equipment is driven by an ultrasonic generator with a frequency range of 18–22 kHz, maximum displacement amplitude of 50 μm and maximum power output of 1 kW. The UIT equipment further contains a piezo-ceramic ultrasonic transducer and an ultrasonic horn made from strength material. The samples of 304 stainless steel are placed in a 0.6 mm groove of the specimen fixture.

### 2.2. Material Properties

According to Yang et al. [61], the elastic limit for nanocrystallized AISI 304 specimens could be enhanced to a level of 42% compared with the untreated specimens. Therefore, stainless steel 304 is chosen for fabricating foam-filled energy absorbing structures. The main chemical composition of this material is shown in Table 1. The nanocrystallized specimens are nanocrystallized on double sides with a processing time 90 s/cm^2^. The tensile specimens are obtained by cutting the nanocrystallized specimens. The material properties of nanocrystallized specimens are measured by tensile test. Universal testing machine AGS-X 300 kN and Epsilon 3442 axial extensometer are utilized for tensile and compression testing. The strain is measured by Epsilon 3442 axial extensometer to ensure test accuracy. The tensile speed is 4 mm/min and the strain–stress relation of tensile specimens is shown in Figure 1. The tensile test result indicates that Young’s modulus of steel is increased from 176.9 GPa to 199.5 GPa after nanocrystallization processing, as depicted in Table 2. Meanwhile, the elastic limit of nanocrystallized specimen reaches 709.6 MP, which has increased by 150%.

Aluminum closed-foam is selected as the filled foam block. The deformation modes of hexagonal foam under quasi-static compression test are shown in Figure 2. To measure its material constants, hexagonal aluminum foam columns with a side length of 26 mm, height of 80 mm and relative density of 0.28 g/cm^3^ are fabricated for the uniaxial compression tests, as shown in Figure 3. The loading speed is 4 mm/min and the stress–strain relation of the foam specimen is presented in Figure 3. Clearly, it has three stages: stage (I), elastic deformation; stage (II), long stable deformation with almost constant stress, defined as the plateau stress *σ_p_*; and stage (III), densification region. From Figure 3, Young’s modulus is 60.4 MPa and the foam plateau stress *σ_p_* is 2.8 MPa [62].

### 2.3. Specimen Preparation

The local nanocrystallized tubes are fabricated by welding two plates. Firstly, two stainless steel plates of SUS 304 are locally nanocrystallized by UIT. Next, each plate is bent to 120° along the bending lines. Then, two bent plates are welded into one thin-walled tube by argon-arc welding. The location for welding lines is in the middle of one side. Lastly, the foam column is filled into the local nanocrystallized tube to form a foam-filled tube, as shown in Figure 4. The geometrical parameters for the regular hexagonal tube are: side length of 26.7 mm, height of 80 mm and thickness of 0.7 mm.

The welding is the last process of the overall manufacturing process. The nanocrystallized steel plate is welded into tubes using the argon-arc welding machine. The welding material shows similar ultimate strength with the 304 steel which was also indicated in the published references [63,64]. In this study, both the nanocrystallized tube and the untreated tube is manufactured by welding. Additionally, the welding lines of both tubes are fabricated using the same welding process. In this situation, the effect of thermal on the comparison of crashworthiness between the untreated tube and nanocrystallized tube is ignorable.

### 2.4. Quasi-Static Axial Compression Tests

The quasi-static compression tests are conducted for untreated empty and foam-filled tubes, local nanocrystallized empty and foam-filled tube and the validity of the test result is verified a finite element analysis. The velocity of load is 4 mm/min during compression, and the whole buckling process is recorded using a digital camera. The compression displacement is set as 70% for the specimen height and all crashworthiness criteria are calculated within the range.

#### 2.4.1. Empty Thin-Walled Tube

As presented in Figure 5, the LNT-2 (Local Nanocrystallized Tube with 2 stripes) load is greater than the UT (untreated tube) load in the whole range. The result in Table 3 indicates that an increase in EA (energy absorption) and SEA (specific energy absorption) of two-stripe tube reaches 46.02% and 46.03% compared to the untreated tube. Meanwhile, PCF (peak crushing force) of the two-stripe tube increases by 13%. The increase for MCF (mean crushing force) reveals that local nanocrystallized tubes have more superior crashworthiness than untreated tubes. Subsequently, the energy absorption capacity for specimens with different stripe numbers is presented in Section 4 and Section 5. Table 3 shows that local nanocrystallization can improve the crashworthiness for thin-walled structures. With regard to deformation energy, the strain energy of nanocrystallized region is increased due to the increase in stress, as shown in Equation (1). The elastic strain energy absorbed by the structure is expressed as:(1)$$W=\frac{1}{2}\sigma_{x}\varepsilon_{x}V,$$
where *W* is the elastic strain energy, σ*_x_* is the normal stress, ε*_x_* is the linear strain and V is the volume of structures.

#### 2.4.2. Foam-Filled Thin-Walled Tube

The compression tests of the FUT (foam-filled untreated tube) and FLNT-2 (foam-filled local nanocrystallized tube) are carried out and Figure 6 shows deformation modes for FUT and FLNT-2 at different compression stages. The load–displacement relation for UT and LNT is shown in Figure 7a,b, respectively. In order to exhibit the interaction effect between the tube and foam, the “LNT-2 + Foam” line is drawn, which denotes the linear summation of crushing force for LNT-2 and foam. The shaded area indicates an increase in energy absorption due to the interaction effect and it increases from 232 J to 445 J, as shown in Table 4. The result shows that local nanocrystallization can dramatically enhance the interaction effect between the wall and foam. Table 4 shows that the specific energy absorption and peak crushing force of FLNT-2 are increased by 40.67% and 28.40% compared to FUT, respectively. However, the increase in peak force has a negative effect on structural crashworthiness. Therefore, it is necessary to optimize the energy capacity of the structures.

## 3. Numerical Simulation

A comprehensive study by FE numerical simulations is conducted using Abaqus/Explicit to analyze, design and optimize local nanocrystallized hexagonal thin-walled tubes. The FE model details are shown in Figure 8. The top of the hexagonal tube is impacted by a moving rigid block with an impact velocity of 3 m/s while the bottom is fixed. The fixed plate and movable wall are defined as rigid components to enhance computational efficiency. The linear elastic theory for an aluminum foam in the elastic stage is adopted and the model of “crushable foam” is used to characterize the material properties in the plastic stage. Finite elements of type are 8-node linear brick (C3D8R) selected in the FE simulation.

Based on the experimental data in Section 2.2, the material properties of the structures are summarized in Table 5 and Table 6, respectively. The classical isotropic plasticity theory of metal is used to define material properties for the UIT-treated area and the untreated area of thin-walled tubes. Small dents are introduced to induce stable deformation modes [62,65,66]. The thin-walled tube is modeled using 4-node shell elements with reduced integration and hourglass control. The surface–surface contact algorithm is adopted to simulate the interaction between the foam and wall based on the real situation. In addition, the general contact between tube and rigid wall is set while the friction coefficient of contact is 0.3. The “penalty contact” is selected in tangential behavior. In the normal behavior, the “Hard Contact” is chosen in this study.

### 3.1. Crashworthiness Assessment

The energy absorption of thin-walled structures is mainly dependent on its deformation during crush. To evaluate the crashworthiness of the structure, several widely used assessment parameters are selected, including energy absorption (EA), specific energy absorption (SEA), peak crushing force (PCF) and mean crushing force (MCF) [54].

EA can be calculated by integrating the displacement force under the load–displacement curve. It is given by
(2)$$EA=\int_{0}^{x_{0}}F(x)dx$$
where *x*_0_ is the crushing displacement and *F(x)* is the crushing force.

*SEA* represents the energy absorption per unit mass and it can accurately measure the energy absorption capacity of the structure. It is given by
(3)$$SEA=\frac{EA}{m}=\frac{\int_{0}^{x_{0}}F(x)dx}{m}$$
where *m* is the total mass of the structure. *MCF* is another effective criterion, which is defined as the average load during the crushing process. *MCF* is expressed as
(4)$$MCF=\frac{\int_{0}^{x_{0}}F(x)dx}{x_{0}}$$

### 3.2. Validation of FE Models

FE models for FUT and FLNT-2 are validated by comparing the deformation modes, load–displacement curve and crashworthiness assessments with an experiment. It is observed from Figure 9 and Figure 10 that numerical prediction for the number of folds, position of the first fold and load–displacement curve is in very agreement with the experiment. Very good agreement between FE solution and the experiment is also observed in Table 7 and the maximum error is less than 10%. From the comparison study, it is concluded that the proposed FE model is accurate and reliable for predicting energy absorption for local nanocrystallized thin-walled hexagonal tubes.

A mesh sensitivity study for aluminum foam is presented in Figure 11a. The mesh size of foam varies from 4 mm to 10 mm. The result shows that the variation in EA is stable [68] for a mesh size smaller than 2 mm. Hence, the mesh size of the foam model is defined as 2.0 mm. Furthermore, a mesh convergence study of shell element is conducted and presented in Figure 11b. It can be seen that EA tends to converge when the mesh size is less than 1.5 mm. Therefore, a mesh size of 1.5 mm is used to model the thin-walled structures in the subsequent numerical cases.

## 4. Result and Discussion

### 4.1. Effects of Local Nanocrystallization Layout on Crashworthiness

The result from the experiment and FE simulation shows that FLNT-2 has a higher SEA compared to FUT. In a previous work [54,69], the authors concluded that local nanocrystallized thin-walled tubes with axial stripes could significantly improve the peak crushing force for buckling-resistant structures and tubes with circumferential horizonal stripes could enhance energy absorption without the expense of an increased peak crushing force. Hence, circumferential continuous horizontal stripes are adopted in this study and the proportion of nanocrystallization is 50%. In addition, empty LNT models with different stripe numbers are simulated to analyze the interaction effect between local nanocrystallization and foam in LNT.

In Figure 12, the final FE deformation modes with different stripe numbers are presented, where light gray denotes the untreated region and blue denotes the nanocrystallized region. In order to confine PCF, the stripe near the top of the hexagonal tube is set as the untreated region. The load–displacement relation of hexagonal tubes with even stripe numbers is presented in Figure 13a, including LNT-2, LNT-4, LNT-6, LNT-8 and LNT-10. The result indicates that all curves of local nanocrystallized specimens are above the curve of the untreated specimen. Figure 13b shows the EA of all LNT structures has a significant enhancement compared with UT. As shown in Figure 14, EA of the hexagonal tube increases first and then decreases with increasing stripe numbers. The crashworthiness assessment is summarized in Table 8. EA of LNT-4 is higher than the other LNT structures with a significant increase in PCF. Additionally, EA of LNT-2 and LNT-6 is essentially equal with values of 997 J and 998 J, respectively. However, PCF of LNT-2 is 29.66 kN, which is lower than that of LNT-6 (37.43 kN). Therefore, LNT-2 is a superior model with reference to all the crashworthiness assessments.

### 4.2. Local Nanocrystallization Layouts on Interaction Effects

The load–displacement relation of FLNTs with different stripes and the corresponding deformation modes are presented in Figure 15a and Figure 16, respectively. It is clear that all FLNTs have better energy absorption performance than FUT, as shown in Figure 15b. The result of FE simulation for FLNTs is summarized in Table 9. It is noted that EA reaches the maximum of 1789 J for FLNT-4, with an increase of 47.12% compared with FUT. Moreover, PCF is only increased by 16.15% compared to the untreated tube. The result reveals that FLNT-4 is a design with superior crashworthiness. As shown in Figure 16, the effect of aluminum foam on the performance is obviously related to the deformation modes and it can be attributed to the interaction effect between the wall and foam. As shown in Figure 17, more foam core in FLNT-4 is extruded into folds compared with FUT, and it dramatically enhances the densification effect. Meanwhile, energy dissipation is contributed by interfacial friction between the foam and tube wall. The correlation between the stripe number and energy absorption of EA and PCF is presented in Figure 18. The model crashworthiness is discounted with increasing stripe numbers. The great discrepancy between energy absorption shows that the crashworthiness of thin-walled structures is significantly affected by local nanocrystallization layouts.

To verify the design parameters of FE numerical simulation, a test specimen for FLNT-4 is fabricated in Figure 19. The deformation modes and load–displacement relation under quasi-static axial loading and crashworthiness criteria are presented in Figure 20 and Table 10, respectively. The experiment result for FLNT-4 shows better energy dissipation performance and the error is less than 5% (compared with FE numerical solution). Therefore, it is concluded that the proposed local surface nanocrystallization design significantly enhances the energy absorption capacity of foam-filled hexagonal thin-walled tubes. 

### 4.3. Foam-Filled Nested Tubular Structure Design

As seen in Table 9, the interaction effect of FLNT-4 is much stronger than that of FUT with an increase of 53.93% of EA. This section aims to explore the strengthening mechanism of energy absorption performance from the aspect of EA and deformation modes.

Table 11 presents the partition energy absorption and contribution of interaction effect for FUT and FLNT-4. The tabular result is obtained by numerical simulation. EA for an individual tube and aluminum foam is also tabulated in Table 11. The interaction effect can be represented by the increase in EA. From the result of tube EA, it is clear that local surface nanocrystallization can sufficiently improve the performance of energy absorption of an individual tube (from 682 J to 1043 J). Meanwhile, it is noted that the tube EA of the nanocrystallization foam-filled tube (FLNT-4, 1264 J) is much larger than that of the nanocrystallization individual tube (1043 J). This observation indicates that local surface nanocrystallization also enhances the interaction effect of foam-filled tubes. Similar observation can be found in the foam EA of foam-filled tubes. It increases from 401 J to 525 J after nanocrystallization (from FUT to FLNT-4). In conclusion, local surface nanocrystallization greatly improves the energy absorption performance of the individual tube as well as the interaction effect between the tube and foam.

The collapse mode for FLNT-4 under axial compression is shown in Figure 21. Clearly, the crushed foam filler can be divided into two regions: (i) densified region and (ii) extremely densified region [25]. The inner part for the foam filler could be approximated into a hexagonal prism, which is extruded as the densified region under axial loading. The surrounding part of foam filler can be approximatively subjected to the multi-axial loading with folding, which results in more and closer interaction between foam cells and nanocrystallization wall. In addition, the interaction effect is generated by additional energy dissipation mechanism, including the resistance of inward bulking and the interfacial friction between the filler and the tube wall [15,42].

Inspired by the above investigation, two nested foam-filled local surface nanocrystallization tubes with four stripes are designed as shown in Figure 22, i.e., (1) N-1-FLNT-4 (Figure 22a), two hexagonal local nanocrystallization tubes with different heights and a ring-like foam core, and (2) N-2-FLNT-4 (Figure 22b), a hexagonal foam filled inside the N-1-FLNT-4. To reduce the initial PCF, the height of the inner tube or the inner hexagonal foam is set to 75 mm, which is 5 mm lower than the outer tube. The crushing force–displacement relations for FLNT-4, N-1-FLNT-4 and N-2-FLNT-4 are illustrated in Figure 23. Clearly, the energy absorption of N-1-FLNT-4 and N-2-FLNT-4 is significantly higher than FLNT-4, which indicates that the nested tube exhibits better energy absorption performance than the single-walled tube. Furthermore, it is observed that the curve of N-2-FLNT-4 has the most peaks, followed by N-1-FLNT-4 and FLNT-4. This phenomenon demonstrates that more plastic folds and foam extrusion behavior are generated in N-2-FLNT-4. To further study the energy absorption performance of the two nested tubes, Figure 24 depicts the deformation modes. Clearly, the aluminum foam in N-2-FLNT-4 is extruded to generate more densified regions than that in N-1-FLNT-4, due to an increase in the contact area between tube wall and foam. Therefore, N-2-FLNT-4 is a superior design. Subsequently, Table 12 presents a comparison of the crashworthiness parameters between N-2-FLNT-4 and FLNT-4. As expected, the EA and SEA of N-2-FLNT-4 are obviously higher than that of FLNT-4, with a 65.51% and 23.99% increase, respectively. It can be concluded that nested foam-filled local surface nanocrystallization structures are shown to be more superior to the single structures in terms of crashworthiness.

## 5. Conclusions

A new type of aluminum foam-filled hexagonal thin-walled structure treated with local nanocrystallization is proposed. The crashworthiness of FLNTs under axial crushing is investigated through an experiment and finite element simulation. The effects of different local nanocrystallization layouts on the energy absorption capacity for FLNTs is studied using numerical simulations. The numerical result concludes that: (i) Proper local surface nanocrystallization can significantly enhance the crashworthiness of hexagonal thin-walled tubes. Compared with UT and FUT, the SEA of LNT-2 is significantly increased by 46.19% and 26.61%, respectively. (ii) FLNT-4 is a better design for aluminum foam-filled local nanocrystallized hexagonal thin-walled tubes with respect to crashworthiness. By comparing the interaction effect for foam-filled tubes, FLNT-4 can dramatically improve the interaction effect from 176 J to 388 J. In addition to numerical analysis, a test specimen for FLNT-4 is fabricated and an experiment is conducted to verify the conclusions. Very good agreement between the experiment and FE numerical solutions has been reported. Moreover, a novel nested foam-filled local surface nanocrystallization tube is designed based on the interaction effect. The simulation result concludes that SEA for N-2-FLNT-4 is further enhanced with an 23.99% increase compared to FLNT-4. The proposed design idea is very promising for the further development of thin-walled energy absorption devices using other metallic and non-metallic materials.

## Figures and Tables

**Figure 1 materials-15-05556-f001:**
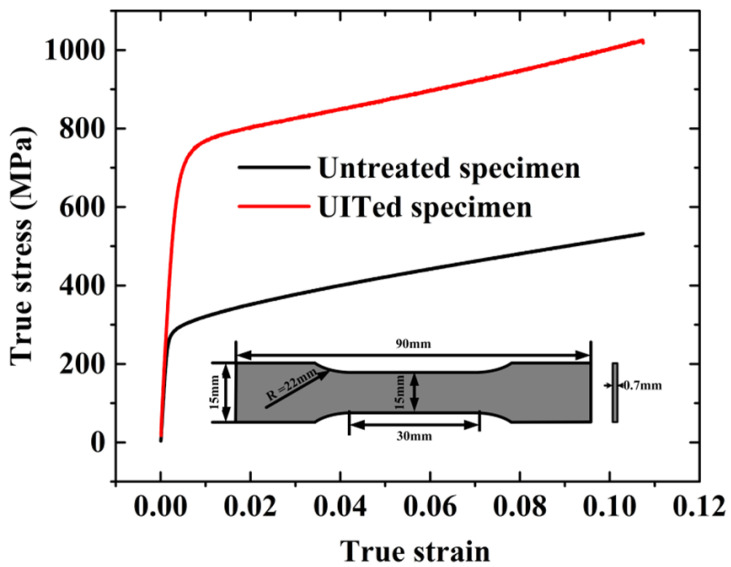
True stress–true strain relation of SUS 304.

**Figure 2 materials-15-05556-f002:**
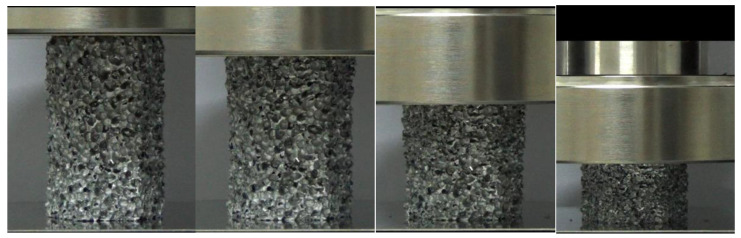
Deformation modes of hexagonal foam under quasi-static compression test.

**Figure 3 materials-15-05556-f003:**
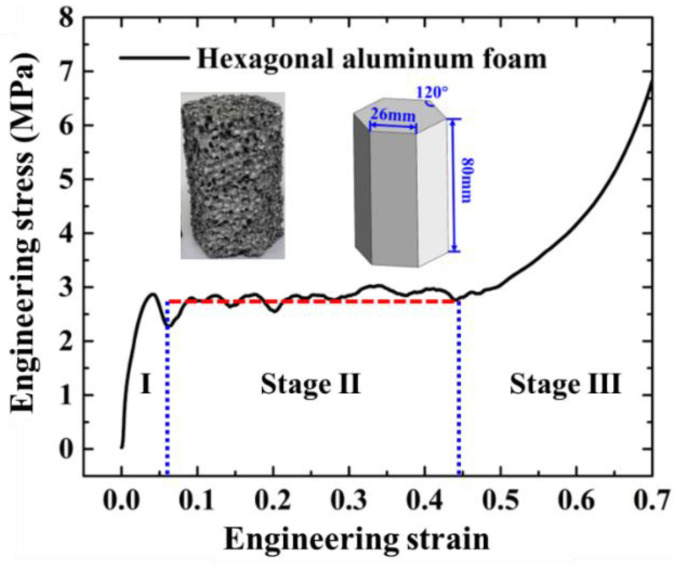
Stress–strain relation of hexagonal aluminum column.

**Figure 4 materials-15-05556-f004:**
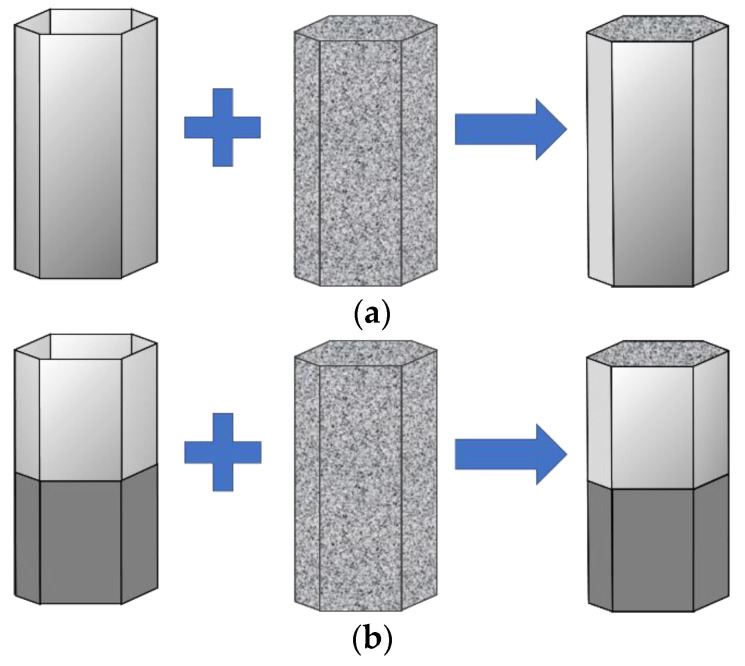
The fabrication process of foam-filled tubes: (**a**) untreated tube; (**b**) local nanocrystallized tube.

**Figure 5 materials-15-05556-f005:**
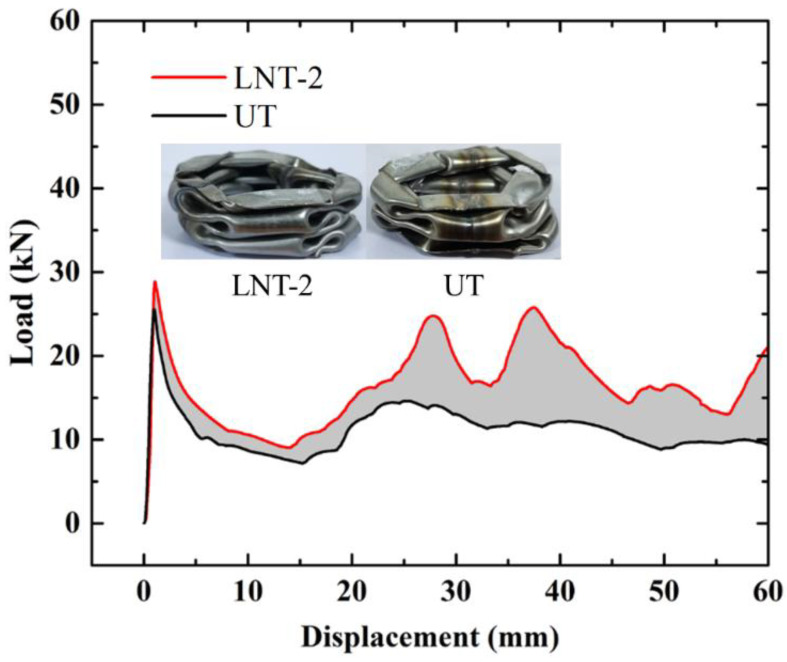
Load–displacement relation of UT and LNT-2.

**Figure 6 materials-15-05556-f006:**
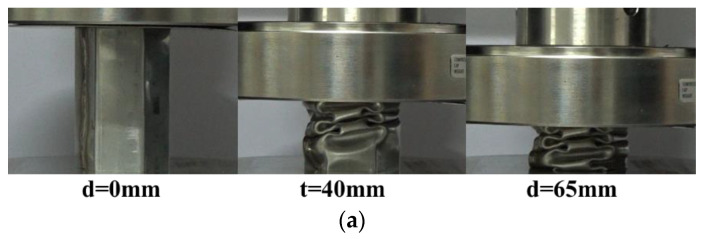
Deformation modes of foam-filled tubes under axial compression test: (**a**) FUT; and (**b**) FLNT-2.

**Figure 7 materials-15-05556-f007:**
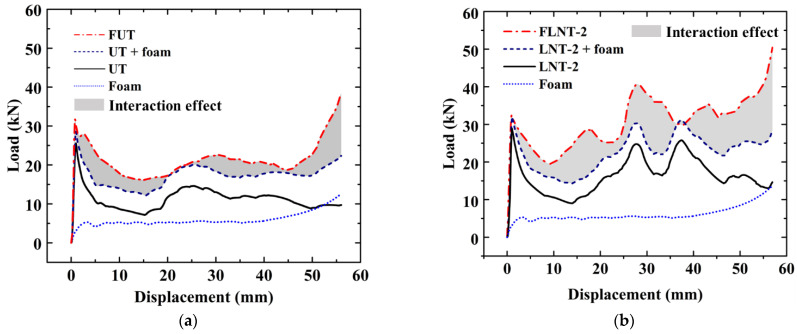
The interaction effect of foam-filled tubes: (**a**) UT and (**b**) LNT-2.

**Figure 8 materials-15-05556-f008:**
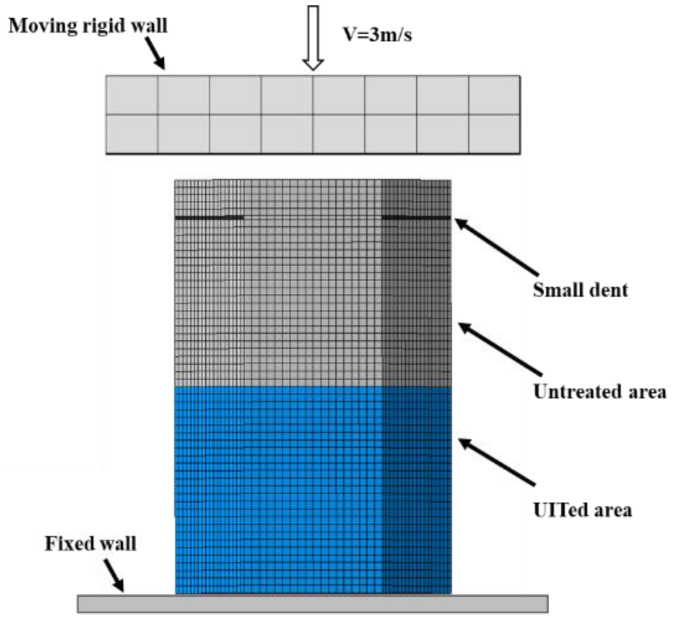
The finite element model.

**Figure 9 materials-15-05556-f009:**
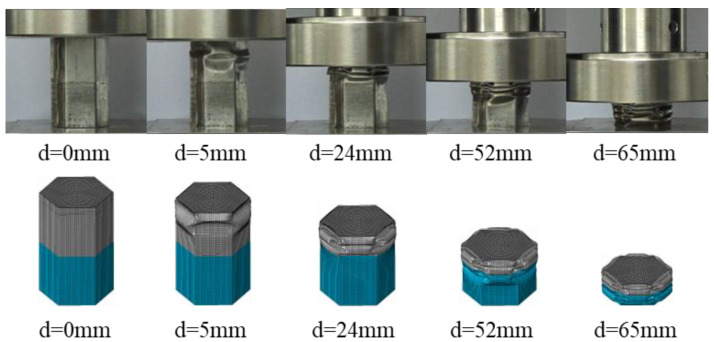
Comparison of two-stripe tube deformation modes by experiment and FE simulation.

**Figure 10 materials-15-05556-f010:**
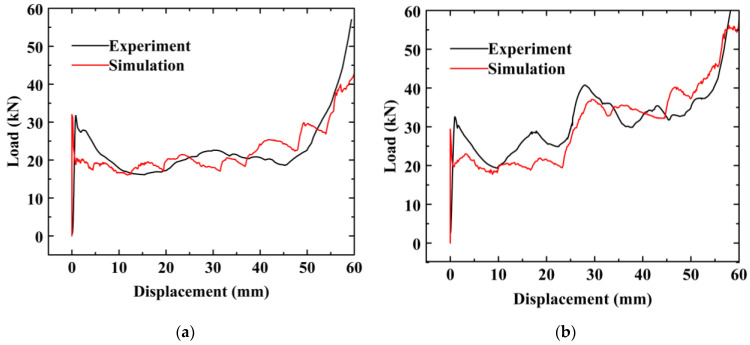
Comparison of load–displacement relation for foam-filled thin-walled tubes: (**a**) FUT and (**b**) FLNT-2.

**Figure 11 materials-15-05556-f011:**
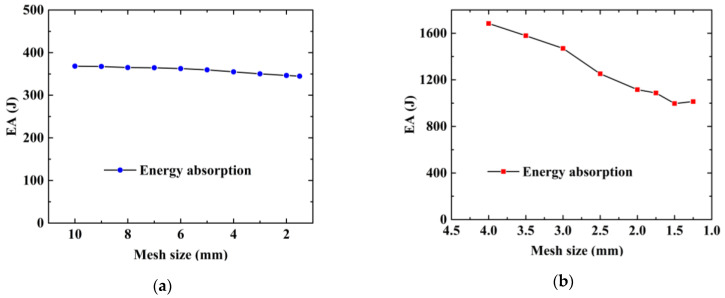
Mesh convergence study: (**a**) aluminum foam and (**b**) local nanocrystallized tube.

**Figure 12 materials-15-05556-f012:**
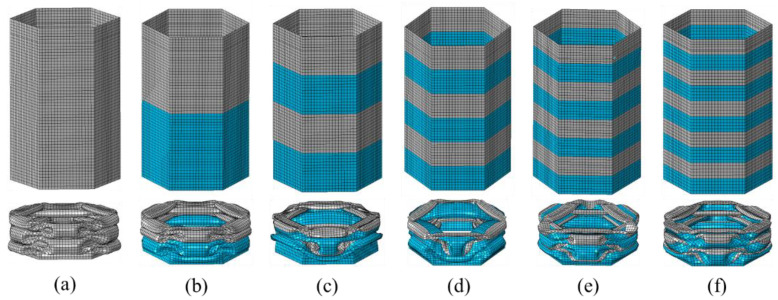
Ultimate deformation modes for models with even circumferential stripes: (**a**) UT; (**b**) LNT-2; (**c**) LNT-4; (**d**) LNT-6; (**e**) LNT-8; (**f**) LNT-10.

**Figure 13 materials-15-05556-f013:**
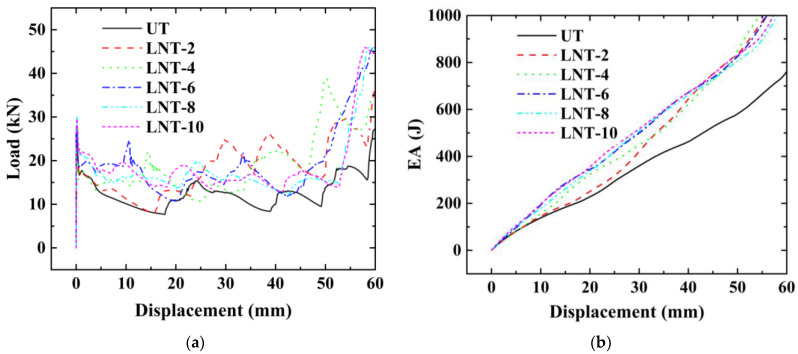
The effect of stripe number for tubes with even circumferential stripes: (**a**) load–displacement relations; (**b**) energy–displacement relations.

**Figure 14 materials-15-05556-f014:**
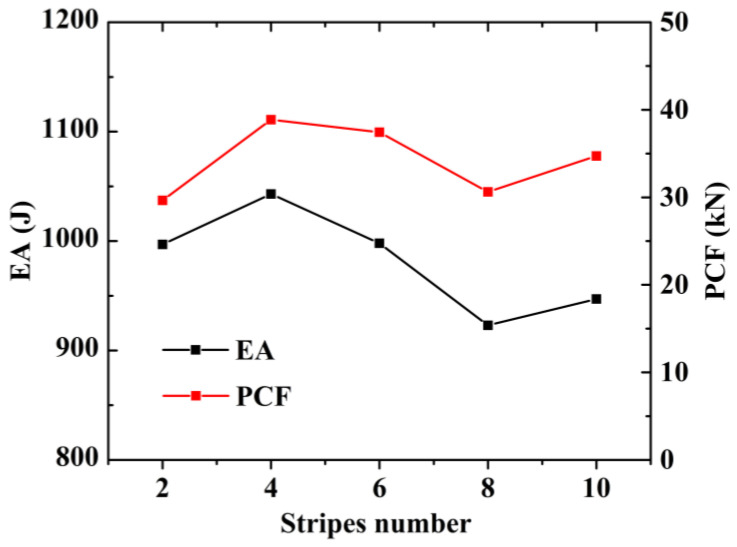
Effect of stripes on energy absorption and PCF.

**Figure 15 materials-15-05556-f015:**
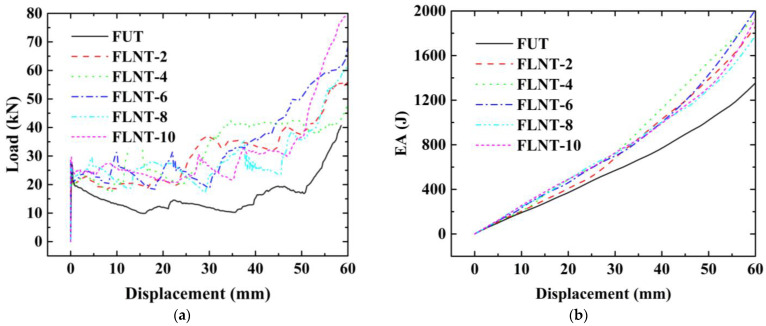
Effect of stripe number for filled tubes: (**a**) load–displacement relations and (**b**) energy–displacement relations.

**Figure 16 materials-15-05556-f016:**
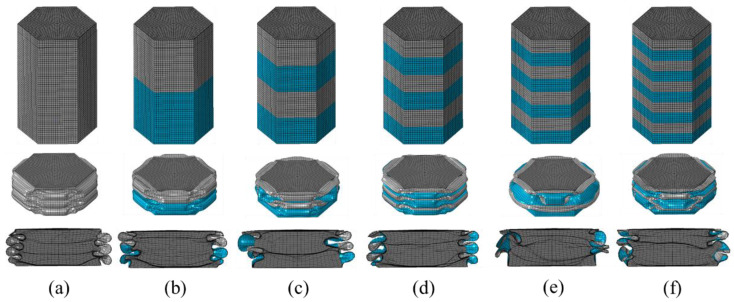
Deformation modes: (**a**) untreated; (**b**) FLNT-2; (**c**) FLNT-4; (**d**) FLNT-6; (**e**) FLNT-8; and (**f**) FLNT-10.

**Figure 17 materials-15-05556-f017:**
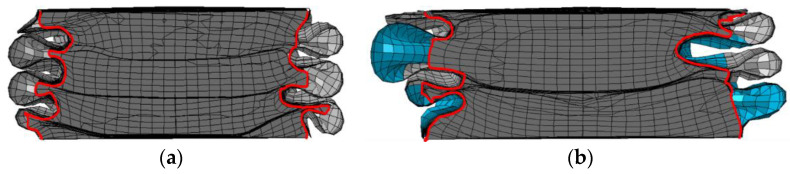
Comparison of densification effect: (**a**) FUT and (**b**) FLNT-4.

**Figure 18 materials-15-05556-f018:**
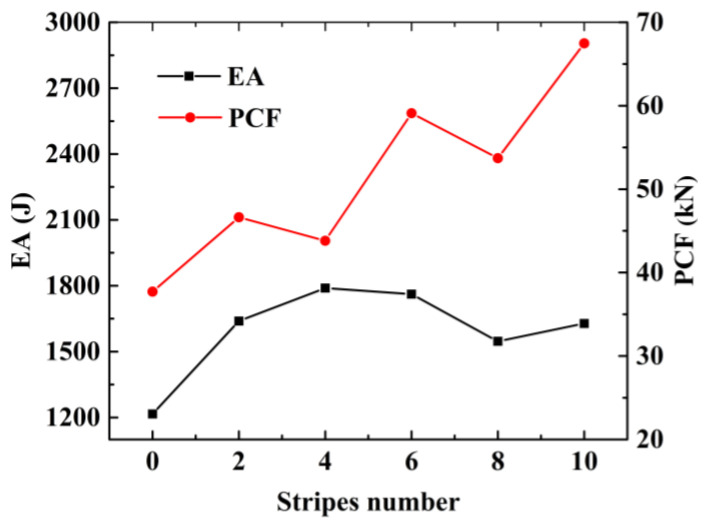
Effect of stripe number on energy absorption and peak crushing force.

**Figure 19 materials-15-05556-f019:**
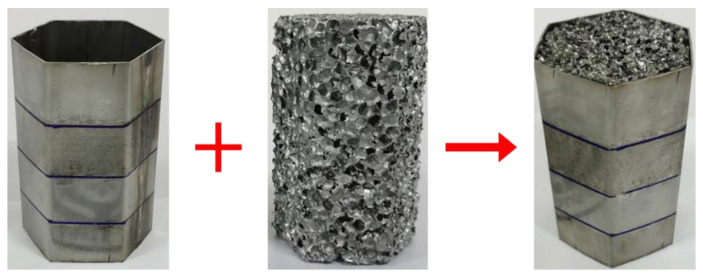
Configuration of FLNT-4.

**Figure 20 materials-15-05556-f020:**
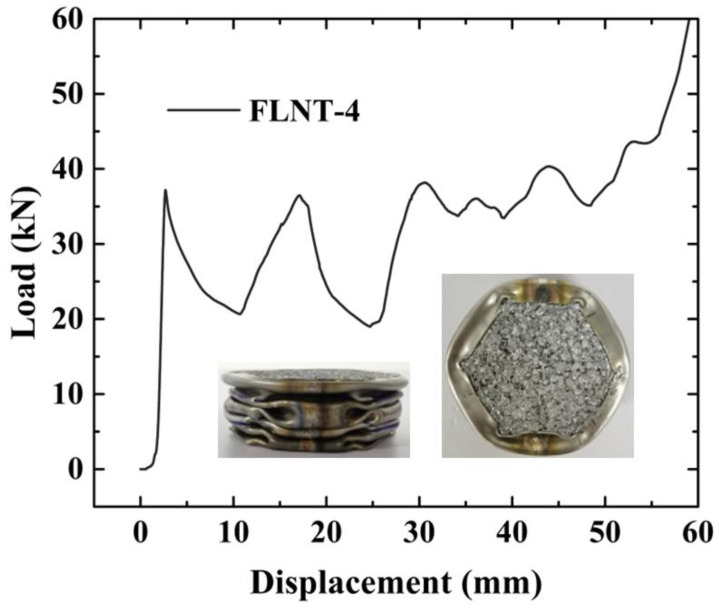
Load–displacement relation for FLNT under compression test.

**Figure 21 materials-15-05556-f021:**
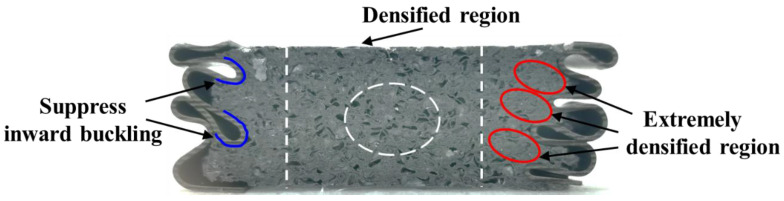
Cut-away image of crushed FLNT-4 section.

**Figure 22 materials-15-05556-f022:**
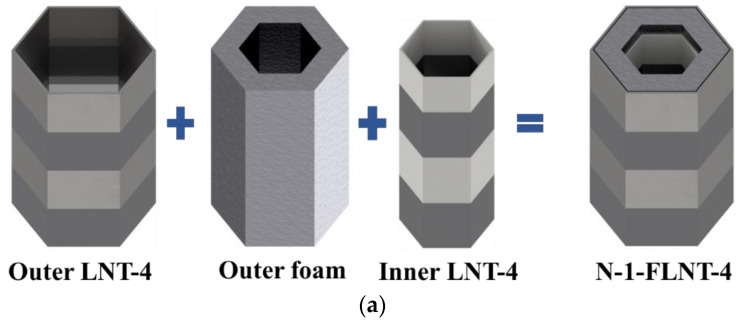
A novel nested foam-filled local surface nanocrystallization tube: (**a**) N-1-FLNT-4; (**b**) N-2-FLNT-4.

**Figure 23 materials-15-05556-f023:**
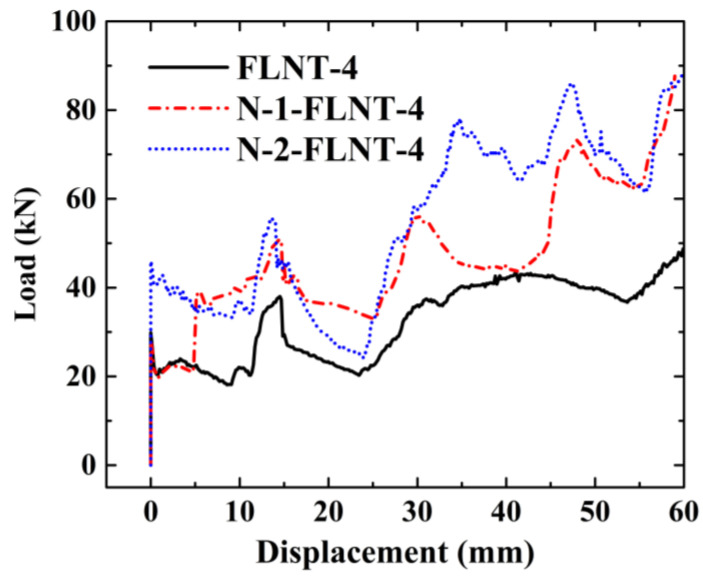
Load–displacement relation of FLNT and N-FLNT-4.

**Figure 24 materials-15-05556-f024:**
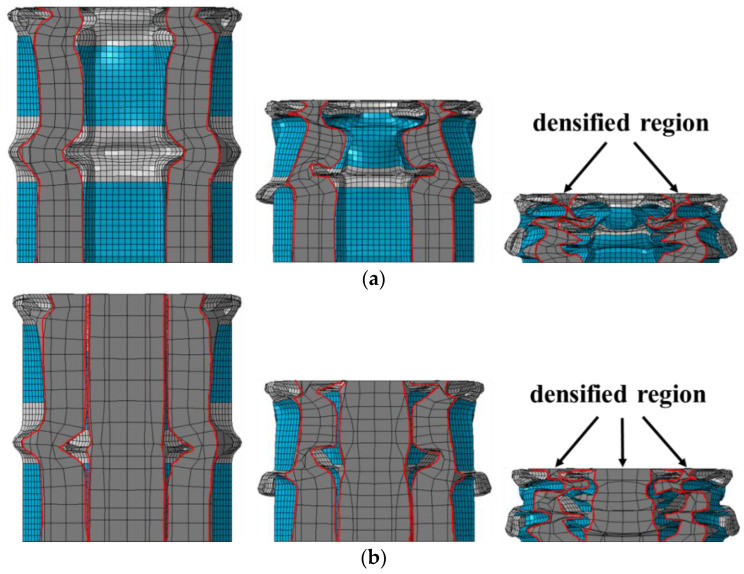
The deformation modes: (**a**) N-1-FLNT-4; (**b**) N-2-FLNT-4.

**Table 1 materials-15-05556-t001:** Chemical compositions of SUS304.

Composition	Fe	Cr	Ni	Mn	Si	N	C	P	S
Percentage	72.06	18.09	8.07	1.29	0.36	0.06	0.04	0.03	0.002

**Table 2 materials-15-05556-t002:** Tensile properties of untreated and nanocrystallized 304 stainless steel.

Tensile Specimens	Young’s Modulus (GPa)	Elastic Limit (MPa)
Untreated	176.9	283.4
Nanocrystallized	199.5	709.6

**Table 3 materials-15-05556-t003:** Energy absorption parameters of empty tubes.

Model	EA (J)	SEA (J/g)	PCF (kN)	MCF (kN)
UT	628.00	10.38	25.55	11.21
LNT-2	917.00	13.01	28.90	16.37
Percentage increase (%)	46.02	25.34	13.11	46.03

**Table 4 materials-15-05556-t004:** Effects of foam on energy absorption for thin-walled tubes.

Type	Model	EA (J)	Mass (g)	SEA (J/g)	PCF (kN)
Foam	Foam	333	42.3	7.87	12.64
Untreated	UT	628	60.5	5.42	25.55
	UT + foam	961	102.8	9.35	28.60
	FUT	1193	110.3	10.82	38.47
	Interaction effect	232	/	/	/
Locally nanocrystallized	LNT-2	917	70.5	13.01	28.90
	LNT2 + foam	1250	112.8	11.08	32.03
	FLNT-2	1695	111.4	15.22	44.90
	Interaction effect	445	/	/	/

**Table 5 materials-15-05556-t005:** Material properties of the thin-walled tube.

Material	ρ (kg/m^3^)	E (GPa)	σ_s_ (MPa)	ν
Untreated	7850	176.9	283.4	0.3
UITed	7850	199.5	709.6	0.3
Source	Experiment	Experiment	Experiment	Experiment

Note: ρ is density of 304 stainless steels, E is Young’s modulus of steels, σ_s_ is elastic limit of steels and ν is Poisson’s ratio.

**Table 6 materials-15-05556-t006:** Material properties for aluminum foam.

Type	ρ_f_ (kg/m^3^)	E_f_ (MPa)	σ_p_ (MPa)	ν_f_	λ
Foam	280	60.4	2.8	0.1	1
Source	Experiment	Experiment	Experiment	Experiment	Ref. [67]

Note: ρ_f_ is density of aluminum foam, E_f_ is Young’s modulus of foam, σ_s_ is elastic limit of foam, ν_f_ is Poisson’s ratio and λ is compression yield stress ratio.

**Table 7 materials-15-05556-t007:** Error analysis of energy absorption between FE solutions and experiment.

Model	Methods	EA (J)	SEA (J/g)	PCF (kN)
FUT	Experiment	1193	10.82	38.47
	Simulation	1216	11.05	37.72
	Error (%)	1.93	2.13	−1.95
FLNT-2	Experiment	1694	15.21	44.76
	Simulation	1639	14.90	46.63
	Error (%)	−3.25	−2.04	4.18

**Table 8 materials-15-05556-t008:** Effect of stripes on energy absorption properties.

Model	EA (J)	SEA (J/g)	PCF (kN)	MCF (kN)
UT	682	9.57	27.13	12.18
LNT-2	997	13.99	29.66	17.80
LNT-4	1043	14.64	38.86	18.63
LNT-6	998	14.01	37.43	17.82
LNT-8	923	12.96	30.61	16.48
LNT-10	947	13.29	34.72	16.91

**Table 9 materials-15-05556-t009:** Crashworthiness data for different stripe number filled tubes.

Model	EA (J)	SEA (J/g)	PCF (kN)	MCF (kN)
FUT	1216	11.05	37.72	21.71
FLNT-2	1639	14.90	46.63	29.27
FLNT-4	1789	16.26	43.81	31.95
FLNT-6	1762	16.02	59.10	31.46
FLNT-8	1547	14.06	53.71	27.63
FLNT-10	1628	14.80	67.48	29.07

**Table 10 materials-15-05556-t010:** Energy absorption properties for FLNT-4.

Model	EA (J)	SEA (J/g)	MCF (kN)	PCF (kN)
Experiment	1730	15.98	30.90	45.56
Simulation	1789	16.26	31.94	43.81
Error (%)	3.41	1.75	3.37	−3.84

**Table 11 materials-15-05556-t011:** Partition energy absorption and contribution of interaction effect via FE analysis.

Model	Configuration	Tube EA (J)	Foam EA (%)	Total EA (J)
FUT	Filled tube components	815	401	1216
	Individuals	682	358	1040
	Increase in EA	133	43	176
FLNT-4	Filled tube components	1264	525	1789
	Individuals	1043	358	1401
	Increase in EA	221	167	388

**Table 12 materials-15-05556-t012:** Energy absorption properties for N-FLNT-4 and FUT.

Model	Mass (g)	EA (J)	SEA (J/g)	PCF (kN)	MCF (kN)
FLNT-4	110	1789	16.26	43.81	31.95
N-2-FLNT-4	147	2961	20.16	85.64	52.88
Percentage increase (%)	33.64	65.51	23.99	95.48	65.51

## Data Availability

This study did not report any data.

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
