# Peer review of "Numerical Simulation and Experimental Study on Energy Absorption of Foam-Filled Local Nanocrystallized Thin-Walled Tubes under Axial Crushing"

_materials, 2022, doi:10.3390/ma15165556_

Round 1

Reviewer 1 Report

- What is the reason to combine thin-walled elements and foam. The idea is know for several decades, but so far no practical application was found. The reason is that the industry can't afford foams. Their cost per kilogram is the same as the cost of pure aluminum. What is the reason behind testing such a combination without the possibility of practical application?

- The language should be significantly improved.

- Please correct the title. It should be a correct English sentence.

- "The samples of 304 stainless steel are placed in a 0.6mm groove of the specimen fixture as shown in Fig 1(a)." The groove and the position of the specimen and the fixture are not visible. Please modify the drawing.

- Nanocrystallization parameters are not well presented for both tensile specimens and crash boxes. Please add the description of the process. It should be improved to allow other researchers to repeat the process.

- In all of the pictures presenting stress-strain curves it is not clear if the authors depict true stress-true strain or engineering stress-engineering strain values.

- The material model for both the steel and the foam should be presented in detail so other researchers could recreate the FEM simulation.

- The clear information about the contacts used in the FE simulation should be presented.

- The mesh size of the foams is unacceptable and way too coarse. It should be refined several times. Then the interaction between the shell and the foam should be visible. The foam will be sucked between the fold of the shell.

Reviewer 2 Report

The manuscript reports development of foam filled local thin walled tubes and its mechanical properties. Experiment and FE analysis is utilized to study the axial crushing of the tubes. The manuscript has some flaws and cannot be accepted as such to the journal. Following are comments on the manscript:

1. The tube material SS304 is nanocrystallized and tensile results are presented in Figure 3. The strength of the material has improved drastically while the ductility remains the same. Explain more on how this is achieved since when strength improves due to nano crystallization, ductility decreases. In addition, the foam filled tube is subjected to compression testing, where as the tube material is studied only for tensile property, given the sheet thickness is 0.7 mm. How do the property can be correlated between the tube and foam filled tube.

2. During specimen preparation it is mentioned the nano crystallized plates were joined to form the tube structure using gas arc welding. Whats the rational of using this approach? Why was a single sheet used to create the tube. Did the welding result in heat affected zone and grain growth in the tube? How did this affect the overall foam filled tube property?

3. Mention how many specimen were tested for mechanical property evaluation. Present results with average and standard deviation. 

4. Too many acronyms are utilized without proper definition. Explains in detail what each acronym stand for (UT, LNT-2, etc.)  

5. How is the foam connected to the outer tube? Comment on this aspect on how the foam interconnected to the outer tube will affect the overall mechanical properties?

6. Why are the results presented in load--displacement instead of stress-strain curves. Since the specimen sizes utilized similar, why are strain curved not presented?

7. Provide suitable references in the manuscrip. What is a Tokyo Yaris energy absorption device? English language needs to be improved in the manuscript

Reviewer 3 Report

The paper deals with energy absorption in foam-filled locally nano-crystallized thin-walled tubes through experiments comparing with numerical simulation. The paper is considered authors’ own work but should be modified with major revision. The followings are the comments for authors.

1.       Page 2, middle part; “Zhang et.al[23]”, Song et.al[24] seems wrong.

2.       Page 2, bottom; “Lu[54] et al.” seems wrong.

3.       Page 3, 3rd line; “Tokyo Yaris” is not possible to understand the meaning.

4.       Table 2; The elastic modulus is not the same for untreated and nano-crystallized samples. Normally, the values should be the same when the sample is with full density. Also, the modulus of 304 stainless steel is known as 190GPa or slightly higher, the reviewer doubt about the data for untreated one. As far as using axial extensometer, the data is not reliable. Other method like ultrasonic tests which is more reliable recommended.

5.       Fig.5; side length is written as 26mm but written in the text as 26.7mm. Which is right?

6.       Page 6, “2.3 Specimen preparation”; It is better to mention that the condition of welding status. It is believed that the area is having more thickness and the microstructure is changed by thermal effect. Is it possible to ignore such effects?

7.       Fig.6; It is better to suggest untreated and nano-crystallized tubes.

8.       Page 7, 6th line; “LNT-2” and “UT” is appeared first and could not understand the materials at this stage.

9.       Equation (1) seems wrong.

10.   Table 3; The same as my comment #8, “EA”, “SEA”, “PCF”, “MCF” are appeared first and could not understand the meaning. Section 3.1 should be appeared earlier.

11.   Page 7, 8th line from bottom; “FUT and FLNT-2” could not understand the meaning

12.   Fig.9; How to draw the line “UT+foamis not explained well. The caption should be considered again because it is not the data only for UT and LMT-2.

13.   Table 5; It is written that the Possion’s ratio is based on experiment. Did the authors measure width or thickness change? It is recommended to be included in the manuscript.

14.   Table 6; Possison’s ratio of aluminum foam is written as 0.1 obtained by experiment. How did the authors measure the value? The reviewer carefully checked Fig.4, and the width is not considered increased. More explanation needed.

Round 2

Reviewer 1 Report

1. This is the corrected title which is still incorrect

“Numerical Simulation and Experiment Study on Axial Crushing Energy Absorption of Foam-filled Local Nanocrystallized Thin-walled Tubes

experiment study? -> experimental study

axial crushing energy absorption?

local -> locally

2. This question was not answered by the authors:

- What is the reason to combine thin-walled elements and foam. The idea is know for several decades, but so far no practical application was found. The reason is that the industry can't afford foams. Their cost per kilogram is the same as the cost of pure aluminum. What is the reason behind testing such a combination without the possibility of practical application?

3. - Using enegineering stress and engineering strain for FEM simullations is not correct.

4. Authors state that the FE model of foam was improved but an old results are still presented in all of the tables as well as in the drawings

Reviewer 2 Report

The manuscript has been revised based on the reviewer comments. However, a few minor modifications are needed and are to be addressed before it can be accepted for publication.

Remove Figures 1 and 2 as these are standard equipments.

Reviewer 3 Report

The paper has been modified most of the parts but please try to modify for the following questions.

1.    On my former question #6, the tube should be welded, so, the thickness of the area should be thicker than the other regular plate. It should be described the sample preparation condition clearly. The reviewer is afraid that the welded part(s) in the tube affects the deformation. Also, it is better to mention in the manuscript that the thermal effect is ignorable.

2.    On my question #5, in the bottom line of page 7, it is still written as “side length 26.7mm”. Is that right?

3.    On my former question #9, the equation (1) seems still strange which suggests elastic strain energy (W).
